Published in FAST Workshop on Smalltalk Related Technologies (11/2022)

# Arrows: a Computer Algebra System in Smalltalk

**Luciano E. Notarfrancesco**                                               *luchiano@gmail.com*

**Reviewed on OpenReview:** *https://openreview.net/forum?id=oIozVmVool*

## Abstract

We describe the design and implementation of a computer algebra system in Smalltalk-80. Here we focus on the categories of rings and modules. We discuss various types of rings and show how to solve systems of linear equations over them, which allows us to construct their categories of finitely presented modules and other interesting objects. We illustrate different parts of the system with concrete examples of use.

## 1 Introduction

The goal of this project is to build an extensible environment where one can construct mathematical objects (such as groups, rings, modules, algebras, schemes, etc) and operate with them, their morphisms and their elements.

The system is built on top of Cuis Smalltalk [Vul22b], an open source and multiplatform Smalltalk-80 system. We try to follow conventional mathematical notation as close as possible, while still adhering to Smalltalk syntax (although we're considering extensions to the Smalltalk syntax as proposed in [CB22, §4 and §5]). Cuis Smalltalk allows Unicode in Smalltalk code [Vul22a], and we take full advantage of this; for example we have messages like $\mathbf{A} \to \mathbf{B}$, $\mathbf{A} \oplus \mathbf{B}$ and $\mathbf{A} \otimes \mathbf{B}$, and global variables like $\mathbb{Z}$ and $\mathbb{Q}$.[1]

The system was built from scratch and it is completely written in Smalltalk without any external dependencies. It currently comprises about 250 classes (which might seem like a lot, but the system still follows strictly Smalltalk's First Design Principle[2] [Ing81] and it is completely comprehensible to a single individual).

Some of the objects implemented include: various types of finite rings and fields, affine algebras, number fields, function fields, finitely presented modules, finite (finitely generated as modules) algebras, schemes (affine schemes, and closed subschemes of affine or projective space), coherent sheaves, and bounded (co)chain complexes in arbitrary abelian categories (e.g. modules, coherent sheaves, or recursively other categories of complexes).

Perhaps surprisingly, many computations with these objects can be reduced to systems of linear equations over some relatively simple commutative ring. The core of the system relies on Gaussian elimination over fields, algorithms for computing Hermite normal forms over Euclidean domains and Howell normal forms over Euclidean rings (possibly with zero divisors), and a generalization of Buchberger's algorithm for computing strong Gröbner bases of modules over polynomial rings with coefficients in Euclidean rings (possibly with zero divisors).

In the following sections we will introduce the fundamental concepts of the system, show examples of use, and go over some constructions and implementation details. We will focus on rings and modules, and some other categories will make a tangential appearance in examples that illustrate how the system works.

---

[1]We can input $\to$, $\oplus$, $\otimes$, $\mathbb{Z}$ and $\mathbb{Q}$ by typing `\to`, `\oplus`, `\otimes`, `\Z` and `\Q` respectively. In general we use standard LaTeX commands to input special characters.

[2]*Personal Mastery: if a system is to serve the creative spirit, it must be entirely comprehensible to a single individual.*

## 2 Domains and Morphisms

The most fundamental concepts in the system are *domains* and *morphisms*. Domains are objects of a category; that is to say, they are the domains and codomains of morphisms.[3] Concrete examples of domains are algebraic structures with an underlying set of elements such as groups, rings and modules, as well as objects without elements such as chain complexes and sheaves.

Given a domain $A$, the message **A id** returns the identity morphism of $A$. And given $B$ in the same category, the message **A $\Rightarrow$ B** answers the hom-set or hom-object $\mathrm{Hom}(A, B)$. We can also compose morphisms $f\colon X \to Y$ and $g\colon Y \to Z$ with the message **g ∘ f**.

The hom message is a functor. Given a morphism $f\colon X \to Y$, **A $\Rightarrow$ f** answers the map from $\mathrm{Hom}(A, X)$ to $\mathrm{Hom}(A, Y)$ that sends a morphism $g$ to $f \circ g$. Similarly, **f $\Rightarrow$ B** answers the map from $\mathrm{Hom}(Y, B)$ to $\mathrm{Hom}(X, B)$ that sends $g$ to $g \circ f$.

### 2.1 Canonical Morphisms

By relying on universal constructions we guarantee that the objects we construct on the computer are, for all intents and purposes, the actual mathematical objects. Universal constructions produce objects equipped with morphisms satisfying certain universal properties, for example a quotient $A/B$ comes with the projection $\pi\colon A \to A/B$. There are also morphisms uniquely determined by their domain and codomain, for example the canonical ring homomorphism from $\mathbb{Z}$ to any (unital) ring.

The message **A $\to$ B** returns the canonical or universal morphism from $A$ to $B$, or **nil** if there's none. This message is a fundamental notion in the system and it is used extensively. For example, the canonical ring homomorphism from $\mathbb{Z}$ to a ring $R$ can be retrieved with the message $\mathbb{Z} \to$ **R**.

For products and other limits there's also **A $\rightrightarrows$ {B₁. ... Bₙ}** that returns a list of projections $\pi_i\colon A \to B_i$, and for coproducts and other colimits **B $\leftleftarrows$ {A₁. ... Aₙ}** returns a list of coprojections $\iota_i\colon A_i \to B$. For example, the projections from a Cartesian product $X := A \times B$ to $A$ and $B$ can be obtained with the message **X $\rightrightarrows$ {A. B}**.

### 2.2 Domains with Elements

Some domains have an underlying set of elements or *carrier set*. Examples of these domains are algebraic structures such as groups, rings and modules, as well as sets themselves. We can test elements for equality **a = b** and membership **a $\in$ A**.

Every element is associated to a *parent* domain, and it is a member of its parent's carrier set. However, an element can be a member of other domains besides its parent, even in different categories. For example, a square matrix has as parent a hom module $\mathrm{End}(R^n)$, but it is also a member of a matrix algebra $\mathcal{M}_n(R)$, and if it is invertible it is also a member of the general linear group $\mathrm{GL}(n, R)$.

Morphisms of domains with elements are *maps* and can be created by specifying the mapping on elements with the message **A to: B map: aBlock**. For example, given a ring $R$ of characteristic $p$, the message **R to: R map: [:x | x ^ p]** answers the Frobenius endomorphism.

There is a coercion mechanism that allows the conversion of elements between domains based on known canonical morphisms. For example, for a unital ring $R$ and a rational integer $n$, we can send the message **R ! n** to obtain $1_R \cdot n$ in $R$ (that is, $1_R + \cdots + 1_R$, or its additive inverse if $n$ is negative). This is equivalent to evaluating the initial morphism of $R$ at $n$ with $\mathbb{Z} \to$ **R value: n**.

It also makes sense to coerce elements between domains of different categories. For example, the number field $\mathbb{Q}(i)$ can be viewed as the vector space $\mathbb{Q}^2$, and while one is a ring and the other is a module, elements of one can be coerced to the other.

---

[3]Unfortunately the name "object" is already taken in Smalltalk. To minimize confusion we avoid saying "domain" to mean "integral domain".

When the carrier set of a domain is countable, we should be able to iterate over its elements without repetitions, with the guarantee that any given element can be reached in a finite number of steps. In some cases the enumeration can be ordered by size.[4]

**Example 2.2.1.** (Enumerating elements). We enumerate the rational integers in order of size (equivalently absolute value, height or bit-size), alternating positive and negative values:

| $\mathbb{Z}$ enumeration

$0, 1, -1, 2, -2, 3, -3, 4, -4, 5, -5, 6, -6, 7, -7, 8, -8, 9, -9, 10, -10 \ldots$

If we can enumerate a ring $R$, we can also enumerate $n$-tuples in $R^n$:

| $(\mathbb{Z}\hat{}2)$ enumeration

$(0,0), (0,1), (1,1), (1,0), (0,-1), (1,-1), (-1,-1), (-1,0), (-1,1), (0,2), (1,2), (-1,2), (2,2), (2,0) \ldots$

For enumerating the positive rationals we use the Calkin-Wilf sequence [CW00]:

$$q_1 := 1; \quad q_{i+1} := \frac{1}{2\lfloor q_i \rfloor - q_i + 1}$$

The enumeration of the full rationals simply adds 0, and after each positive rational $q_i$ adds $-q_i$. The Calkin-Wilf sequence enumerates the rationals in order of increasing height (or equivalently bit-size).

| $\mathbb{Q}$ enumeration

$0, 1, -1, 1/2, -1/2, 2, -2, 1/3, -1/3, 3/2, -3/2, 2/3, -2/3, 3, -3, 1/4, -1/4, 4/3, -4/3, 3/5, -3/5 \ldots$

For some rings we also implement enumeration of ideals, maximal ideals, prime ideals, primes, irreducibles, etc. For example we enumerate the prime ideals of $\mathbb{Z}$:

| $\mathbb{Z}$ primeIdeals

$0, 2\mathbb{Z}, 3\mathbb{Z}, 5\mathbb{Z}, 7\mathbb{Z}, 11\mathbb{Z}, 13\mathbb{Z}, 17\mathbb{Z}, 19\mathbb{Z}, 23\mathbb{Z}, 29\mathbb{Z}, 31\mathbb{Z}, 37\mathbb{Z}, 41\mathbb{Z}, 43\mathbb{Z}, 47\mathbb{Z}, 53\mathbb{Z}, 59\mathbb{Z}, 61\mathbb{Z}, 67\mathbb{Z}, 71\mathbb{Z} \ldots$

And the primes of the Gaussian integers $\mathbb{Z}[i]$:

| GaussianField new integers primes

$1+i, 1-i, -1-i, -1+i, 2+i, 2-i, -2+i, -2-i, 3i, 1+3i, -1+3i, 2+3i, -2+3i, 3, 1-3i, -1-3i \ldots$

And we also support other messages from the Collection protocol, implemented as possibly infinite enumerations:

| $\mathbb{Q}$ select: [:x | x numerator odd]

$1, -1, 1/2, -1/2, 1/3, -1/3, 3/2, -3/2, 3, -3, 1/4, -1/4, 3/5, -3/5, 5/2, -5/2, 5/3, -5/3, 3/4, -3/4, 1/5 \ldots$

| $\mathbb{Q}$ collect: [:x | x^2]

$0, 1, 1/4, 4, 1/9, 9/4, 4/9, 9, 1/16, 16/9, 9/25, 25/4, 4/25, 25/9, 9/16, 16, 1/25, 25/16, 16/49, 49/9, 9/64 \ldots$

We can also generate elements at random. This is straightforward when the carrier set is finite, and otherwise it is usually possible to sample elements at random from a finite subset of elements of bounded size.

**Example 2.2.2.** (Generating elements at random). When the carrier set of an object $A$ is finite, we can sample elements uniformly at random using the message **A atRandom**:

| $(\mathbb{Z}/4)$ atRandom

2

| $(\mathbb{Z}/2\hat{}8)$ atRandom

---

[4]For some notion of *size*, for example the number of bits required by their internal representation in the computer's memory, or possibly by other more intrinsic notions of size such as degree, arithmetic height, Euclidean length, etc.

> 01101011

| ($\mathbb{Z}/12$^2) automorphisms atRandom

$$\begin{bmatrix} 1 & 0 \\ 7 & 5 \end{bmatrix}$$

But if the carrier set of $A$ is not finite **A atRandom** cannot generate elements with a uniform distribution. In that case, if we want to sample elements uniformly we need to specify a finite subset to sample elements from, for example the subset of elements that can be stored in the computer memory with at most a given number of bits, polynomials of at most a given degree, etc:

| $\mathbb{Z}$ atRandomBits: 10

> $-410$

| ($\mathbb{Z}/4$) polynomials atRandomMaxDegree: 5

> $2x^4 + x^2 + 1$

## 3   Rings and Modules

We model the category of unital rings. We require rings to be associative, but not necessarily commutative.

In order to do anything with rings in a computer we need to at least be able to compute the basic arithmetic operations of addition, multiplication, and additive inverse, as well as testing equality of elements (or equivalently testing if an element is zero). Additionally, ring elements must implement:

- **a lift: b**  answers $x$ such that $ax = b$ if such an $x$ exists, and **nil** otherwise;
- **a colift: b**  answers $x$ such that $xa = b$ if such an $x$ exists, and **nil** otherwise.[5]

We implement left division **b \ a** and right division **a / b** in terms of lift and colift respectively, producing an error when a solution doesn't exist. The lift and colift operations also allow us to compute multiplicative inverses and test if an element is a unit. Note that when $a$ is a zero divisor $ax = b$ and $xa = b$ don't have unique solutions, and in that case the lift and colift operations can return *any* solution.

### 3.1   Involutive Rings

We say that a ring $R$ is an *involutive* ring (or a $*$-ring) if it is equipped with an antiautomorphism $* \colon R \to R$ that is also an involution: $(ab)^* = b^*a^*$ and $(a^*)^* = a$. In particular, commutative rings are involutive with the trivial involution $a^* = a$. Matrix algebras are also involutive with the involution given by matrix transposition $A^* = A^\mathsf{T}$.

Elements of involutive rings are expected to implement **a opposite** to answer $a^*$. This allows to implement **a colift: b** in terms of the lift as **(a opposite lift: b opposite) opposite**.

### 3.2   Euclidean Rings

Following Samuel in [Sam71], we say that a commutative ring $R$ is *Euclidean* if it is equipped with a function $\varphi \colon R \to W$ to a well-ordered set $W$ and an algorithm that given $a, b \in R, b \neq 0$ computes $q, r$ with $a = bq + r$ and $\varphi(r) < \varphi(b)$. With this definition a Euclidean ring is not required to be an integral domain (it can contain nontrivial zero divisors), and thus it includes rings like $\mathbb{Z}/m\mathbb{Z}$, $\mathbb{Z} \times \mathbb{Z}$ and Galois rings.

Elements of Euclidean rings are expected to implement the following basic operations:

- **a // b**  answers the quotient $q$ of the Euclidean division $a = bq + r$ with $\varphi(r) < \varphi(b)$;
- **a gauge**  answers $\varphi(a)$;
- **a normalization**  answers a unit $u$ such that $au$ is a unique choice of associate;
- **a annihilator**  answers a generator of the annihilator ideal $\mathrm{Ann}(a) = \{x \in R : ax = 0\}$.

---

[5]The reason behind the names *lift* and *colift* will become clear in §3.3 when we talk about computable rings.

The remainder **a \\ b** of the Euclidean division is simply computed as **a − (a // b · b)**.[6] The lift and colift operations are also implemented in terms of Euclidean division. And via the Euclidean algorithm, Euclidean rings support the messages **a gcd: b** and **a xgcd: b** (that returns both the GCD and the Bézout coefficients).

In terms of the normalization we implement **a normalized** as **a · a normalization**. This gives us a unique choice of associate for each element, generalizing the idea of taking absolute value of a rational integer, making a polynomial monic or echelonizing a matrix. And in terms of the annihilator we implement also **a isZeroDivisor** as **a annihilator isZero not**. Note also that if we have a method to generate elements of a ring $R$ at random, the normalization message allows us to generate elements of the group of units $R^{\times}$ at random simply by taking an element of $R$ at random and returning its normalization unit.

Algorithms that work over Euclidean rings, such as algorithms for matrix normal forms and Gröbner bases, are implemented in a most general form, allowing arbitrary rings as long as they have the required properties and support the required basic operations.

### 3.3 Computable Rings

We say that a ring $R$ is *computable* if there are algorithms for solving homogeneous and inhomogeneous linear systems over $R$, which amounts to the following problems on matrices:

- *syzygies problem*: given a matrix $A$, find $X$ such that $AX = 0$, and $X$ is universal in the sense that for any $Y$ with $AY = 0$ there exists $T$ with $XT = Y$;
- *lifting problem*: given matrices $A$ and $B$, decide whether there exists a matrix $X$ such that $AX = B$, and in the affirmative case construct such matrix.

In other words, the syzygies problem asks for a matrix $X$ whose columns generate the module of syzygies of the columns of $A$. And, identifying matrices acting on the left with homomorphisms of free modules, the lifting problem asks for a lift of $B$ along $A$ making the following diagram commute:

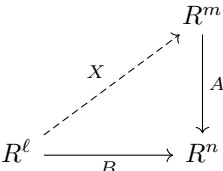

In addition we define *cosyzygies* and *colifting* with the multiplication reversed, corresponding to the linear systems $XA = 0$ and $XA = B$.

We implement the syzygies and lifting problems on matrices as messages **A syzygies** and **A lift: B**, and cosyzygies and colifting as **A cosyzygies** and **A colift: B**. And in analogy with left and right division of ring elements we implement **B \ A** and **A / B** for matrices in terms of lift and colift respectively, producing an error when the solution doesn't exist. Note that when the matrices belong to the algebra $\mathcal{M}_n(R)$ of square matrices over a commutative ring $R$, the lift, colift, left division and right division messages just defined on matrices are consistent with their counterparts on ring elements.[7]

When the ring of coefficients is commutative we can compute **A cosyzygies** as **A transpose syzygies transpose** and **A colift: B** as **(A transpose lift: B transpose) transpose**. More generally, over involutive rings we compute cosyzygies as **A opposite syzygies opposite** and colift as **(A opposite lift: B opposite) opposite**, where the message **A opposite** answers the *conjugate transpose* of $A$, i.e. the transpose of the matrix obtained by applying the involution of $R$ to each coefficient of $A$. Note that the ring $\mathcal{M}_n(R)$ of square matrices over an involutive ring $R$ is itself an involutive ring with **A opposite** as its involution.

---

[6]It's unfortunate to have an apparent inconsistency between the message **a \\ b** which is standard in Smalltalk-80 for the remainder of the Euclidean division, and the message **b \ a** introduced by us for left division. However, this is not really much of a problem because we implement Euclidean division only in commutative rings, where we don't need to distinguish between left and right division and just use **a / b**.

[7]Moreover, when $R$ is a PIR the syzygies message is equivalent to the annihilator, and when $R$ is a PID the *coechelonization* matrix (that is, the unimodular matrix $U$ such that $AU$ is in column echelon form) is equivalent to the normalization unit defined for ring elements.

It's straightforward to solve linear systems over fields using Gaussian elimination. Over Euclidean domains we can solve linear systems using the Hermite normal form.[8] And over general Euclidean rings we can solve linear systems using the Howell normal form (see [How86], [SM98] and [Sto00]). Thus, Euclidean rings are computable.

Furthermore, computability turns out to be a very stable property, preserved by products, quotients, localizations and finite extensions. In the following sections we will elaborate on some of these constructions.

### 3.4 Modules

Given a ring $R$, we would like to construct ideals of $R$, and more generally $R$-modules. For this, we implement (unital) finitely presented right $R$-modules.[9] Objects and morphisms in the category of finitely presented $R$-modules can be internally represented as matrices with coefficients in $R$. More precisely, we represent a module $M$ with a *relations matrix* and a *generator matrix* corresponding to the arrows of a free presentation $R^m \to R^n \to M$. And if we have another module $N$ given by a presentation $R^{m'} \to R^{n'} \to N$, we represent a module homomorphism $M \to N$ with a $n' \times n$ matrix (we identify matrices with homomorphisms between free modules, and more general module homomorphisms are not matrices but are internally represented by a matrix). For details on these constructions see [GP07], [BR08], [BLH11].

Most computations that one would want to do with finitely presented $R$-modules and their homomorphisms can be reduced to solving linear systems over $R$.[10] In particular, if $R$ is computable, *all the abelian category axioms for finitely presented R-modules are constructive*, as established by Barakat and Lange-Hegermann in [BLH11], and this means we can develop effective homological algebra on them.

**Example 3.4.1.** (Free modules, tuples, matrices). We implement free modules $R^n$ as modules of $n$-tuples with the canonical basis:

| (ℤ^3) basis

$((1, 0, 0), (0, 1, 0), (0, 0, 1))$

We identify homomorphisms of free modules with matrices acting on the left on tuples. When $R$ is commutative we implement $\operatorname{Hom}(R^n, R^m)$ as the $R$-module of $m \times n$ matrices:

| ℤ^3 ⇒ (ℤ^2)

$\mathbb{Z}^{2 \times 3}$

| ℤ^3 ⇒ (ℤ^2) ! #(1 2 3 4 5 6)

$\begin{bmatrix} 1 & 2 & 3 \\ 4 & 5 & 6 \end{bmatrix}$

Now, if we want the unique module homomorphism from the trivial module $\mathbb{Z}^0$ to $\mathbb{Z}^3$ we can simply do:

| ℤ^0 → (ℤ^3)

$0$

This is printed as 0 because its a zero morphism, but it is actually a $3 \times 0$ matrix.

Moreover, we identify the direct sum of free modules $R^n \oplus R^m$ with the free module $R^{n+m}$.

| ℤ^2 ⊕ (ℤ^3)

$\mathbb{Z}^5$

---

[8]For efficiency we use specialized algorithms in some cases, for example over $\mathbb{Z}$ we use Dixon's $p$-adic expansion method [Dix82].

[9]We don't need to implement left modules explicitly, as they are equivalent to right modules over the opposite ring. In particular, when $R$ is an involutive ring, $R^{\operatorname{op}}$ can be identified with $R$ via the involution.

[10]One important problem that cannot be reduced to solving linear systems is the *module isomorphism problem*: given two $R$-modules, decide whether they are isomorphic and produce an isomorphism. This can be solved in some rings using matrix normal forms, and there are other techniques for when $R$ is a finite-dimensional algebra.

In order for this identification to make sense, we ensure that $R^{n+m}$ is always equipped with the corresponding projections and coprojections, for example:

```
| ℤ^5 ⇉ {ℤ^2. ℤ^3}
```

$$\left( \begin{bmatrix} 1 & 0 & 0 & 0 & 0 \\ 0 & 1 & 0 & 0 & 0 \end{bmatrix}, \begin{bmatrix} 0 & 0 & 1 & 0 & 0 \\ 0 & 0 & 0 & 1 & 0 \\ 0 & 0 & 0 & 0 & 1 \end{bmatrix} \right)$$

The message **A** ⊕ **B** applied to matrices $A$ and $B$ answers the block-diagonal matrix with $A$ on the top-left and $B$ on the bottom-right. When seeing the matrices as homomorphisms of free modules, this matches the direct sum of homomorphisms.

And we also have messages **A** ⊓ **B** to get the matrix whose rows are the rows of $A$ followed by the rows of $B$, and **A** ⊔ **B** to get the matrix whose columns are the columns of $A$ followed by the columns of $B$. These messages are implemented also for general module homomorphisms, and correspond to the product and coproduct morphisms in additive categories [BLH11, §2 (8), (9)].

**Example 3.4.2.** (Finitely generated abelian groups). Abelian groups are the same as $\mathbb{Z}$-modules. To be precise, there is not only an equivalence of categories but also an isomorphism of categories, so they are *the same* in the strongest sense. At the level of Smalltalk this manifests naturally as *polymorphism* and allows us to identify finitely generated abelian groups with finitely generated $\mathbb{Z}$-modules. More concretely, this means that group homomorphisms can have $\mathbb{Z}$-modules as domain or codomain and can be composed with homomorphisms of $\mathbb{Z}$-modules.

We can construct $\mathbb{Z}$-modules from other commutative groups. For example here we construct a $\mathbb{Z}$-module from a permutation group. We use the message **A span: aSet** to create the subobject of $A$ generated by a set of elements, in particular here we create a subgroup of $S_6$ generated by an element given by images: the permutation that sends 1 to 6, 2 to 1, 3 to 2, etc. Note that permutations are printed as a list of cycles, in this case just one cycle (4 3 2 1 6 5):

```
| S := SymmetricGroup new: 6
```

$S_6$

```
| H := S span: {#(6 1 2 3 4 5)}
```

$\langle(4\ 3\ 2\ 1\ 6\ 5)\rangle$

```
| G := H asAbelianGroup
```

$\mathbb{Z}^2/\langle(2,0);(0,3)\rangle$

```
| G isCyclic
```

true

```
| G isTorsion
```

true

Here we create an abelian group from a product of rings:

```
| G := (ℤ × (ℤ/3) × (ℤ/5) × (ℤ/12)) asAbelianGroup
```

$\mathbb{Z}^4/\langle(0,3,0,0);(0,0,5,0);(0,0,0,12)\rangle$

```
| G invariants
```

$(0,3,60)$

```
| G primaryInvariants
```

$\{0,5,4,3^2\}$

```
| G torsion
```

$\mathbb{Z}^2/\langle(3,0);(0,60)\rangle$

| G torsion exponent

60

**Example 3.4.3.** (Pullbacks and pushouts). We implement *pullback* and *pushout* of module homomorphisms with the messages **f** ∧ **g** and **f** ∨ **g**. Given two morphisms $f\colon X \to Z$ and $g\colon Y \to Z$, the pullback **f** ∧ **g** is an object $P$ and two morphisms $\pi_1\colon P \to X$ and $\pi_2\colon P \to Y$ such that the following diagram commutes:[11]

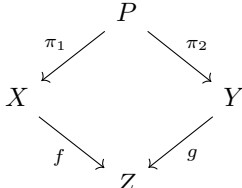

and it is universal with respect to this diagram, meaning that for any other such $P'$, $\pi_1'$ and $\pi_2'$, there exists a unique $u\colon P' \to P$ such that $\pi_1 \circ u = \pi_1'$ and $\pi_2 \circ u = \pi_2'$.

We compute $P$ as **(f ⊔ g negated) kernel**, and equip it with the two projections. (The pushout is dual to the pullback, it's the same with the arrows reversed and cokernel instead of kernel.)

Here we show how we use the pullback to compute intersection of submodules. First we construct two submodules $S, T \subset \mathbb{Z}^3$ with nontrivial intersection.

```
M := ℤ^3.
a := M atRandom
```

$(-5, 5, -1)$

```
S := M span: {M atRandom. a}
```

$\langle(1,1,0);(0,10,-1)\rangle$

```
T := M span: {M atRandom. a}
```

$\langle(5,-5,0);(0,0,1)\rangle$

Then we compute the intersection $S \cap T$ as a pullback. We take the inclusions of the two submodules into their ambient module, compute their pullback, and then return the image of any of the projections $\pi_1$ or $\pi_2$ (of course we also have the message **S** ∩ **T** that does exactly this):

```
((S → M) ∧ (T → M) ⇉ {S. T}) anyone image
```

$\langle(5,-5,1)\rangle$

This ends up performing a computation similar to the Zassenhaus algorithm [Zas66], but we express it in an abstract and more general form without any mention of elements, coefficients or matrices. Note that, although in this example we worked with submodules of a free $\mathbb{Z}$-module for simplicity, *this works for finitely presented modules over any computable ring.*

**Example 3.4.4.** (Betti numbers of the Klein bottle). Once we have constructed modules as abelian categories, we can construct chain complexes of modules and compute homology. We use this to compute the Betti numbers of the Klein bottle.

First we build a simplicial complex from a minimal triangulation of the Klein bottle:

---

[11]There are several different notations for the pullback and the pushout. We particularly like the symbol ∧ for the pullback because it matches the upper part of the pullback diagram, i.e. the output of the message **f** ∧ **g**. Similarly, if we write arrows going down, the symbol ∨ matches the lower part of the pushout diagram, i.e. the output of the message **f** ∨ **g**.

```
X := SimplicialComplex facets: #(
    (2 3 7) (1 2 3) (1 3 5) (1 5 7)
    (1 4 7) (2 4 6) (1 2 6) (1 6 0)
    (1 4 0) (2 4 0) (3 4 7) (3 4 6)
    (3 5 6) (5 6 0) (2 5 0) (2 5 7))
```

Then we construct its chain complex over $\mathbb{Z}$:

```
C := X chainComplexOver: ℤ
```

$$0 \leftarrow \mathbb{Z}^8 \leftarrow \mathbb{Z}^{24} \leftarrow \mathbb{Z}^{16} \leftarrow 0$$

And then we compute the Betti numbers of the chain complex (that is, the ranks of the homology groups):

```
C betti
```

$(1, 1, 0)$

### 3.5   Product Rings

Given rings $R_1, \ldots, R_n$, the direct product ring $R := R_1 \times \cdots \times R_n$ can be constructed as the set of tuples $(r_1, \ldots, r_n)$ with $r_i \in R_i$ and multiplication and addition defined component-wise. The product ring $R$ comes equipped with projections $\pi_i \colon R \to R_i$, and a composition function that takes a list of elements of the components $r_i \in R_i$ to an element of the product $R$.

We can solve linear systems over $R$ by projecting to the components $R_i$, solving the projected systems separately, and then composing the solutions back to obtain solutions over $R$.

Note that if all the components $R_i$ are Euclidean, then the product ring $R$ is also Euclidean [Sam71, Proposition 6], and in that case we could alternatively solve linear systems by means of the Howell normal form.

It is worth mentioning that we construct the special case of the power product of rings $R^{\times n} = R \times \cdots \times R$ as a finite $R$-algebra (see § 3.8.2), taking advantage of optimizations described in § 4.

**Example 3.5.1.** (Chinese remainder theorem). We can decompose $\mathbb{Z}/m\mathbb{Z}$ as a product of local rings $\mathbb{Z}/p_1^{e_1}\mathbb{Z} \times \cdots \times \mathbb{Z}/p_\ell^{e_\ell}\mathbb{Z}$, where $m = p_1^{e_1} \cdots p_\ell^{e_\ell}$ is the prime factorization of $m$.

```
R := ℤ/60
```

$\mathbb{Z}/60\mathbb{Z}$

```
P := R decomposition
```

$\mathbb{F}_5 \times \mathbb{F}_3 \times \mathbb{Z}/2^2\mathbb{Z}$

When we construct this product ring we equip it with ring homomorphisms that allow coercion of elements in both directions.

```
a := R atRandom
```

33

```
b := P ! a
```

$(3, 0, 1)$

```
R ! b
```

33

**Example 3.5.2.** (A finite ring). We perform some basic computations with the finite ring $\mathbb{Z}/4\mathbb{Z} \times \mathbb{Z}/4\mathbb{Z}$.

```
R := ℤ/4 × (ℤ/4)
```

$\mathbb{Z}/2^2\mathbb{Z} \times \mathbb{Z}/2^2\mathbb{Z}$

The ring has 16 elements (12 zero divisors and 4 units):

| R zeroDivisors asSet

$\{10, 12, 30, 32, 00, 01, 02, 20, 03, 21, 22, 23\}$

| R units asSet

$\{31, 13, 33, 11\}$

The Jacobson radical is nontrivial:

| R radical

$\langle 22 \rangle$

The ring contains two maximal ideals, so it is not a local ring:

| R maximalIdeals

$\{\langle 12 \rangle, \langle 21 \rangle\}$

And it contains three nonzero nilpotents:

| R nilradical asSet

$\{00, 22, 20, 02\}$

### 3.6 Quotient Rings

Given a commutative ring $R$ and a finitely generated ideal $I$, we construct the quotient ring $R/I$ equipped with the canonical projection $\pi \colon R \to R/I$.

An element $\bar{a}$ of the quotient ring $R/I$ is a residue class and can be represented internally as an element $a \in R$ (a *representative* of the residue class $\bar{a} = a + I$), defining equality as $\bar{a} = \bar{b} \iff a - b \in I$.

Linear systems over $R/I$ can be reduced to linear systems over $R$. Suppose $I = \langle f_1, \ldots, f_\ell \rangle$. For solving the syzygies problem $\bar{A}\bar{X} = 0$ over $R/I$ we construct the matrix $A$ whose coefficients are representatives of the coefficients of $\bar{A}$ and solve the following system over $R$:

$$\left[ \begin{array}{c|ccccc} & f_1 & & & f_\ell & \\ A & & \ddots & \cdots & & \ddots \\ & & & f_1 & & & f_\ell \end{array} \right] X = 0$$

If $\bar{A}$ has $n$ rows, the solution $\bar{X}$ we're looking for is the matrix whose coefficients are the projections to $R/I$ of the coefficients of the first $n$ rows of $X$. And we solve the lifting problem $\bar{A}\bar{X} = \bar{B}$ in exactly the same way, by reducing to a lifting problem over $R$.

As with product rings, it is worth noting that quotients of Euclidean rings are also Euclidean, so when $R$ is Euclidean we could alternatively solve linear systems over $R/I$ by means of the Howell normal form.

In the case that $R/I$ is known to be a field (that is, $I$ is known to be a maximal ideal), we can solve linear systems over $R/I$ with Gaussian elimination. We do this in fact when $R/I = \mathbb{Z}/p\mathbb{Z}$.

**Example 3.6.1.** (Modular integers). The ring $\mathbb{Z}/m\mathbb{Z}$ of residue classes of integers modulo $m$ is an example of a quotient ring. When $m$ is a power of a prime $p$ we have the local ring $\mathbb{Z}/p^k\mathbb{Z}$ with maximal ideal $\langle p \rangle$, and when $k = 1$ the prime field $\mathbb{F}_p = \mathbb{Z}/p\mathbb{Z}$.

| $\mathbb{Z}/12$

$\mathbb{Z}/12\mathbb{Z}$

| $\mathbb{Z}/16$

$\mathbb{Z}/2^4\mathbb{Z}$

> $\mathbb{Z}/17$

$\mathbb{F}_{17}$

When $n$ divides $m$ we have a canonical ring homomorphism $\mathbb{Z}/m\mathbb{Z} \to \mathbb{Z}/n\mathbb{Z}$. In particular, we have canonical homomorphisms $\pi_i \colon \mathbb{Z}/p^k\mathbb{Z} \to \mathbb{Z}/p^i\mathbb{Z}$ for all $1 \le i \le k$.

> $\mathbb{Z}/16 \to (\mathbb{Z}/8)$

$\pi \colon \mathbb{Z}/2^4\mathbb{Z} \to \mathbb{Z}/2^3\mathbb{Z}$

This allows us to form a projective system:

$$\mathbb{F}_p \leftarrow \mathbb{Z}/p^2\mathbb{Z} \leftarrow \mathbb{Z}/p^3\mathbb{Z} \leftarrow \cdots$$

The $p$-adic integers $\mathbb{Z}_p$ are the inverse limit of this system, and we have projections $\pi_k \colon \mathbb{Z}_p \to \mathbb{Z}/p^k\mathbb{Z}$ for each $k \ge 1$.

As other computer algebra systems, we implement $p$-adic integers $\mathbb{Z}_p$ as approximations with fixed precision. However, this is not rigorous enough because completions like $\mathbb{Z}_p$ are not really computable (we can't even test elements for equality). A more rigorous alternative is to work explicitly in a prime power ring $\mathbb{Z}/p^k\mathbb{Z}$ for a big enough $k$.[12] That's one of the reasons why the rings $\mathbb{Z}/p^k\mathbb{Z}$ are of computational importance and we distinguish them from the more general rings $\mathbb{Z}/m\mathbb{Z}$. Another important application of $\mathbb{Z}/p^k\mathbb{Z}$ is the construction of Galois rings (see Example 3.8.1).

**Example 3.6.2.** (Linear codes). Other interesting objects we can construct from modules are linear codes. Originally linear codes were considered mostly over fields, but in the early 1990s Hammons et al. [HKC$^+$94] discovered that several remarkable nonlinear binary codes are actually linear codes over $\mathbb{Z}/4\mathbb{Z}$, and linear codes over finite rings quickly became a hot topic in coding theory. This discovery also sparkled a resurgence of interest in Galois rings (see Example 3.8.1).

This is the octacode $O_8$ over $\mathbb{Z}/4\mathbb{Z}$ [Wan97, Example 1.3].

> ```
> M := Z/4^8 span: #(
>     (1 0 0 0 3 1 2 1)
>     (0 1 0 0 1 2 3 1)
>     (0 0 1 0 3 3 3 2)
>     (0 0 0 1 2 3 1 1))
> ```

$\langle 10003121; 01001231; 00103332; 00012311 \rangle$

> ```
> C := M asLinearCode
> ```

$[8, 4, 4]_4$

This code has 256 codewords, minimum Hamming distance 4, and it is self-dual:

> ```
> C size
> ```

256

> ```
> C minimumDistance
> ```

4

> ```
> C = C dual
> ```

true

(See Example 3.8.6 for a linear code over a noncommutative ring.)

---

[12]Similarly, instead of working with the power series ring $R[\![x_1, \ldots, x_n]\!]$, we can work in $R[x_1, \ldots, x_n]/\langle x_1, \ldots, x_n \rangle^k$ for a big enough $k$.

### 3.7    Localizations

Given a commutative ring $R$ and a multiplicatively closed subset $S \subset R$ we can construct $S^{-1}R$, the localization of $R$ at $S$. The elements of this ring are pairs $(a, b)$ noted as fractions $a/b$ with $a \in R$ and $b \in S$, with equality defined as $a/b = c/d \iff (ad - cb)s = 0$ for some $s \in S$. This construction comes equipped with the localization morphism that sends an element $a \in R$ to $a/1$.

One important type of localization is the localization at a prime ideal. Given a prime ideal $\mathfrak{p}$ of $R$, we set $S := R \setminus \mathfrak{p}$ and define the localization of $R$ at $\mathfrak{p}$ as $R_{\mathfrak{p}} := S^{-1}R$. This is a local ring and its maximal ideal is $\mathfrak{p}R_{\mathfrak{p}}$ (the extension of $\mathfrak{p}$ by the localization map).

We also construct $\mathrm{Frac}(R)$, the total ring of fractions of $R$, as the localization at the multiplicative set consisting of all the regular elements of $R$ (that is, elements that are not zero divisors). The elements of $\mathrm{Frac}(R)$ are fractions $a/b$ with $a, b \in R$ and $b$ regular. When $R$ is an integral domain $\mathrm{Frac}(R)$ is a field, and there's also a canonical morphism from $R_{\mathfrak{p}}$ to $\mathrm{Frac}(R)$.[13]

Conditions for the computability of general localizations $S^{-1}R$ were established by Posur in [Pos18]. In particular, linear systems over $R_{\mathfrak{p}}$ and $\mathrm{Frac}(R)$ can be reduced to linear systems over $R$.

Note that with this approach we can compute in localizations of affine algebras without having to deal with standard bases and local monomial orderings. See § 3.8.1 for more details.

As with products and quotients, localizations of Euclidean rings are also Euclidean [Sam71, Proposition 7], so when $R$ is Euclidean we could alternatively solve linear systems over localizations $S^{-1}R$ by means of the Howell normal form.

When $R$ is an integral domain $\mathrm{Frac}(R)$ is a field and we can also solve linear systems over $\mathrm{Frac}(R)$ with Gaussian elimination (although in general this is very inefficient).

**Example 3.7.1.** (Localizations of $\mathbb{Z}$ and $\mathbb{Q}[x]$). The total ring of fractions of $\mathbb{Z}$ is $\mathbb{Q}$:

```
ℤ fractions
```

$$\mathbb{Q}$$

Localizations of Dedekind domains at nontrivial prime ideals are discrete valuation rings (that is, local PIDs that are not fields).

```
p := ℤ · 5.
L := ℤ @ p
```

$$\mathbb{Z}_{\langle 5 \rangle}$$

```
L maximalIdeal
```

$$\langle 5 \rangle$$

The elements of $\mathbb{Z}_{\langle 5 \rangle}$ are reduced fractions $a/b$ where $b$ is not multiple of 5, and the maximum power of 5 that divides $a$ is the *valuation* of $a/b$:

```
a := L ! (50 / 29).
a valuation
```

$$2$$

Moreover, we have unique factorization:

```
a factorization
```

$$\{5^2\}$$

Similarly for $\mathbb{Q}[x]$:

---

[13]For both constructions $R_{\mathfrak{p}}$ and $\mathrm{Frac}(R)$ we require $R$ to have a GCD algorithm that we use to automatically reduce numerators and denominators, otherwise computations can quickly become impractical.

```
R := ℚ polynomials.
R fractions
```

$$\mathbb{Q}(x)$$

```
x := R x.
p := R · (x^2 + 1).
L := R @ p
```

$$\mathbb{Q}[x]_{\langle x^2+1 \rangle}$$

```
f := L ! ((x^2 + 1)^3 · (x - 2)) / (L ! (x^2 + x))
```

$$(x^7 - 2x^6 + 3x^5 - 6x^4 + 3x^3 - 6x^2 + x - 2)/(x^2 + x)$$

```
f valuation
```

3

```
f factorization
```

$$\{(x^2 + 1)^3\}$$

**Example 3.7.2.** (Localizations of finite rings). We don't need to construct all localizations explicitly as rings of formal fractions. Localizations of $\mathbb{Z}/m\mathbb{Z}$ are isomorphic to $\mathbb{Z}/n\mathbb{Z}$ for some $n$ dividing $m$, and in this case the localization map is (and cannot be other than) the canonical projection $\mathbb{Z}/m\mathbb{Z} \to \mathbb{Z}/n\mathbb{Z}$, which is not injective unless $n = m$.

```
R := ℤ/60.
p := R · 3.
R @ p
```

$$\mathbb{F}_3$$

```
p := R · 2.
L := R @ p
```

$$\mathbb{Z}/2^2\mathbb{Z}$$

We see that the localization map takes every element not in $\mathfrak{p}$ to a unit in $R_{\mathfrak{p}}$:

```
φ := R → L
```

a RingMap: $\mathbb{Z}/60\mathbb{Z} \to \mathbb{Z}/2^2\mathbb{Z}$

```
R allSatisfy: [:x | x ∈ p or: [(φ value: x) isUnit]]
```

true

Moreover, since in a finite ring $R$ an element is either a unit or a zero divisor, the total ring of fractions $\mathrm{Frac}(R)$ is $R$ itself:

```
R fractions
```

$$\mathbb{Z}/60\mathbb{Z}$$

## 3.8  Algebras

Given a commutative ring $R$, we construct some subcategories of $R$-algebras. We require them to be associative and unital so they are also rings (under our definition of ring). We have $R$-algebra homomorphisms between them, and since we're viewing them in the category of rings we also allow ring homomorphisms that are not necessarily $R$-algebra homomorphisms.

$R$-algebras come equipped with the inclusion ring homomorphism from $R$.

### 3.8.1 Affine Algebras

We refer to polynomial rings and their quotients as *affine algebras* because they are the coordinate rings of affine varieties. Any finitely generated commutative associative algebra is isomorphic to an affine algebra.

We construct a polynomial ring $R[x_1, \ldots, x_n]$ from a commutative ring $R$ and the free abelian monoid $[x_1, \ldots, x_n]*$ consisting of monomials $x_1^{e_1} \cdots x_n^{e_n}$. The monoid comes equipped with an ordering that is used when computing Gröbner bases. Polynomials are simply finite formal linear combinations of monomials with coefficients in $R$. This construction is a particular case of a *monoid algebra*.

As we saw in § 3.6, we can reduce linear systems over an affine algebra $R[x_1, \ldots, x_n]/I$ to linear systems over the polynomial ring $R[x_1, \ldots, x_n]$. And linear systems over a polynomial ring $R[x_1, \ldots, x_n]$ can be solved using Gröbner bases techniques. We implement a generalization of Buchberger's algorithm that computes strong Gröbner bases over $R[x_1, \ldots, x_n]$ for any Euclidean ring $R$ (possibly with nontrivial zero divisors), based on [EPP21] and [EH21].

In addition to Gröbner bases with global monomial orderings, the algorithm can also compute standard bases with local monomial orderings using Mora's normal form [Mor82; Mor91]. This is the classic approach to compute in localizations of affine algebras [GP07, Example 1.5.3 (4)]; but instead of doing that we reduce to linear systems over the affine algebra as explained in § 3.7, which bypasses completely the need to work with Mora's normal form and local monomial orderings and seems to be more efficient [BLH11, § 4.4] [Pos18, Construction 4.3].

**Example 3.8.1.** (Galois rings). We construct number fields, function fields, Galois rings and Galois fields as affine algebras. For example, we construct the Galois field $\mathbb{F}_{p^r}$ as the quotient $\mathbb{F}_p[x]/\langle f \rangle$ for some irreducible monic polynomial $f \in \mathbb{F}_p[x]$ of degree $r$.

| F := GaloisField new: 3 to: 2

$\mathbb{F}_{3^2}$

We see that in this case $\mathbb{F}_{3^2}$ was constructed as $\mathbb{F}_3[x]/\langle x^2 - x - 1 \rangle$:

| F cover

$\mathbb{F}_3[x]$

| F relations

$\langle x^2 - x - 1 \rangle$

More generally, we construct the Galois ring $\mathrm{GR}(p^k, r)$ of characteristic $p^k$ and order $p^{kr}$ as the quotient $\mathbb{Z}/p^k\mathbb{Z}[x]/\langle f \rangle$ for a monic polynomial $f \in \mathbb{Z}/p^k\mathbb{Z}[x]$ of degree $r$ such that the reduction of $f$ modulo $p$ is irreducible in $\mathbb{F}_p[x]$. When $k = 1$ this is just the Galois field $\mathbb{F}_{p^r}$. Galois rings have very nice properties, for example they are local rings with maximal ideal $\langle p \rangle$ (which is also the set of zero divisors), and they are Euclidean rings. For details on Galois rings see [GM73], [McD74] and [BF02].

| R := GaloisRing new: 3 to: 3 to: 2

$\mathrm{GR}(3^3, 2)$

The ideals of $\mathrm{GR}(p^k, r)$ are of the form $\langle p^i \rangle$ for $0 \leq i \leq k$ and form a chain $\langle p^{i+1} \rangle \subset \langle p^i \rangle$.

| R ideals

$0, \mathrm{GR}(3^3, 2), \langle 3 \rangle, \langle 9 \rangle$

And given $\mathrm{GR}(p^k, r)$, there's a canonical projection to $\mathrm{GR}(p^\ell, r)$ for all $\ell \leq k$ with kernel $\langle p^\ell \rangle$. This accounts for all ideals.

| S := GaloisRing new: 3 to: 2 to: 2

$\mathrm{GR}(3^2, 2)$

| (R $\rightarrow$ S) kernel

⟨9⟩

| (R → F) kernel

⟨3⟩

The generalization from Galois fields to Galois rings was very easy to implement, and thinking of Galois fields $\mathbb{F}_{p^r}$ and prime power rings $\mathbb{Z}/p^k\mathbb{Z}$ uniformly as special cases of Galois rings made much of our code simpler (see for example how we treat Galois fields and Galois rings uniformly at the lowest level in §4).

**Example 3.8.2.** (An affine curve). We can construct closed subschemes of affine or projective space. Here's a simple example with an affine plane curve.

```
P := ℚ polynomialsIn: #(x y).
x := P x: 1.  y := P x: 2.
R := P / (x^2 + (x^3) - (y^2)).
X := R spec
```

Spec $\mathbb{Q}[x,y]/\langle x^3 + x^2 - y^2 \rangle$

The following figure shows the curve $x^3 + x^2 - y^2 = 0$ in blue, and the tangent cone $x = y$ and $x = -y$ in red:

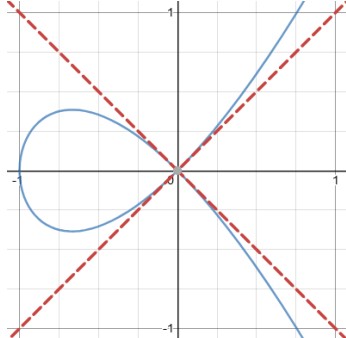

As we see in the figure, this curve is singular at the origin:

```
p := X ! (0,0).
p isSingular
```

true

The tangent cone at the origin is the union of the two lines $x = y$ and $x = -y$, and the tangent space is the whole plane:

```
p tangentCone
```

Spec $\mathbb{Q}[x,y]/\langle x^2 - y^2 \rangle$

```
p tangentSpace
```

$\mathbb{A}^2(\mathbb{Q})$

**Example 3.8.3.** (Projective lines over finite rings). Following [SPKP07, §2], we explore the projective line over $\mathbb{Z}/4\mathbb{Z} \times \mathbb{Z}/4\mathbb{Z}$.

```
R := ℤ/4 × (ℤ/4).
P := (R polynomialsIn: 2) proj
```

$\mathbb{P}(\mathbb{Z}/2^2\mathbb{Z} \times \mathbb{Z}/2^2\mathbb{Z})$

We see that the line has 36 rational points:

```
S := P points asSet.
S size
```

36

We partition points in two distinct groups: type I when at least one of the coordinates is a unit, and type II when both coordinates are zero divisors. We see that there are 28 points of type I:

```
| S count: [:p | p coordinates anySatisfy: [:any | any isUnit]]
```

28

For any finite commutative ring, the number of points of type I is always equal to the sum of the order of the ring plus the number of zero divisors:

```
| R size + R zeroDivisors size
```

28

Moreover, we say that two points $[a{:}b]$ and $[c{:}d]$ are *neighbors* or *parallel*, and note it $[a{:}b] \parallel [c{:}d]$, if $ad - cb$ is not a unit. We say that two points are *distant* if they are not neighbors, and we call *neighborhood* the set of all neighbors of a given point.

We take 3 pairwise distant points and compute their neighborhoods:

```
| u := P points ! #(1 0).
| v := P points ! #(0 1).
| w := P points ! #(1 1).
| Nᵤ := S select: [:x | x ∥ u]
```

$\{[12{:}01], [11{:}21], [11{:}22], [11{:}23], [11{:}30], [11{:}32], [12{:}21], [21{:}10], [21{:}12], [01{:}10],$
$[01{:}12], [10{:}01], [10{:}21], [11{:}00], [11{:}01], [11{:}02], [11{:}03], [11{:}10], [11{:}12], [11{:}20]\}$

```
| Nᵥ := S select: [:x | x ∥ v]
```

$\{[21{:}10], [12{:}01], [12{:}11], [12{:}21], [12{:}31], [20{:}11], [21{:}11], [21{:}12], [21{:}13], [22{:}11],$
$[00{:}11], [01{:}10], [01{:}11], [01{:}12], [01{:}13], [02{:}11], [10{:}01], [10{:}11], [10{:}21], [10{:}31]\}$

```
| Nw := S select: [:x | x ∥ w]
```

$\{[11{:}33], [11{:}21], [11{:}23], [11{:}30], [11{:}31], [11{:}32], [12{:}11], [12{:}31], [21{:}11], [21{:}13],$
$[01{:}11], [01{:}13], [10{:}11], [10{:}31], [11{:}01], [11{:}03], [11{:}10], [11{:}11], [11{:}12], [11{:}13]\}$

We see that each neighborhood contains 20 points, their triple intersection is empty, and the pairwise intersections contain 8 points each. We also see that each neighborhood contains 4 Jacobson points (that is, points unique to that particular neighborhood).

The following is a schematic sketch of the structure of $\mathbb{P}(\mathbb{Z}/4\mathbb{Z} \times \mathbb{Z}/4\mathbb{Z})$ taken from [SPKP07].

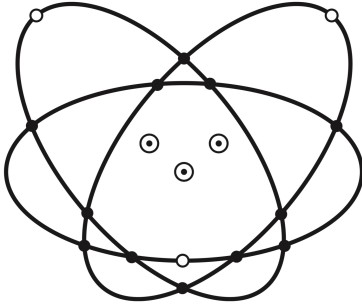

Each small circle or bullet represents two distinct points. The three double circles represent the pairwise distant points $u$, $v$ and $w$, and their neighborhoods are represented by ellipses centered on each one of them. The three white circles represent Jacobson points.

**Example 3.8.4.** (Hilbert series). We can compute Hilbert series of affine algebras. First we compute the Hilbert series of $\mathbb{Q}[x, y, z, w]$:

```
R := ℚ polynomialsIn: #(x y z w).
H := R hilbertSeries
```

$$1/(1 - 4t + 6t^2 - 4t^3 + t^4)$$

If we factorize the denominator we see that $H = (1 - t)^{-4}$:

```
H denominator factorization
```

$$\{(t - 1)^4\}$$

Now we look at a quotient of this polynomial ring.

```
x := R x: 1. y := R x: 2. z := R x: 3. w := R x: 4.
f1 := x·z - (y^2).
f2 := x·w - (y·z).
f3 := y·w - (z^2).
I := R · {f1. f2. f3}.
```

$$\langle y^2 - xz, yz - xw, z^2 - yw \rangle$$

```
H := (R/I) hilbertSeries
```

$$(1 + 2t)/(1 - 2t + t^2)$$

The order of the pole at 1 is the Krull dimension of $R/I$:

```
(H orderAt: 1) negated
```

2

And we see that in fact it matches the Krull dimension, which is computed by other means:

```
(R/I) dimension
```

2

### 3.8.2 Finite Algebras

Given a finitely generated $R$-module over a commutative ring $R$, we can construct an $R$-algebra by explicitly giving a bilinear map that defines the multiplication. This is called a finite $R$-algebra, or a finite-dimensional $R$-algebra if $R$ is a field.

We model the associative unital case in the category of rings, and the general case (not necessarily associative or unital, such as Lie algebras and octonions) in a separate category of *distributive* algebras.

Some examples of finite associative $R$-algebras currently implemented include rings of integers of number fields, Clifford algebras, quaternion algebras, group algebras of (very) finite groups, and algebras of module endomorphisms (in particular matrix algebras). Also we can construct all finite rings as finite $\mathbb{Z}$-algebras with a finite abelian group as underlying module [CT16].

Solving linear systems over a finite $R$-algebra can be reduced to solving linear systems over $R$ via the regular representation. And in the particular case that the finite algebra is a division ring, we can also solve linear systems by Gaussian elimination, since the Gaussian elimination algorithm doesn't require the coefficients ring to be commutative.

**Example 3.8.5.** (Quaternion algebras). We implement the quaternion algebra $\mathbb{H}_{a,b}(R)$ over a commutative ring $R$ as a finite $R$-algebra with underlying module $R^4$ and basis $\{1, i, j, k\}$, with multiplication defined such that $i^2 = a$, $j^2 = b$ and $ij = -ji = k$, where $a$ and $b$ are units of $R$ (the *invariants* of the quaternion algebra).

```
H := QuaternionAlgebra over: ℤ invariants: #(1 1)
```

$$\mathbb{H}_{1,1}(\mathbb{Z})$$

```
H basis
```

$(1, i, j, k)$

We can coerce elements between the algebra and its underlying module:

| a := H atRandom

$1 - i - j - 4k$

| H asModule ! a

$(1, -1, -1, -4)$

We can compute the (left) regular representation map of $a$, i.e. the module endomorphism that sends the module element corresponding to $x$ to the module element corresponding to $ax$ (we have the message **a representation** to do this, but here we are explicit for illustrative purposes):

| A := H asModule endomorphisms map: [:x | H asModule ! (a · (H ! x))]

$$\begin{bmatrix} 1 & -1 & -1 & 4 \\ -1 & 1 & -4 & 1 \\ -1 & 4 & 1 & -1 \\ -4 & 1 & -1 & 1 \end{bmatrix}$$

We generate another element at random and take its coordinates:

| b := H atRandom

$1 + j + 2k$

| v := H asModule ! b

$(1, 0, 1, 2)$

And we see that the coordinates of $ab$ are given by $Av$:

| A · v

$(8, -3, -2, -3)$

| a · b

$8 - 3i - 2j - 3k$

The regular representation is fundamental for many computations with finite algebras. For example we use it to compute lift, annihilator, characteristic and minimal polynomials, norm, trace, and to reduce linear systems over a finite $R$-algebra to linear systems over $R$.

**Example 3.8.6.** (Linear codes over noncommutative rings). Linear codes can also be constructed over noncommutative rings. Here we construct the linear 3-quasi-cyclic rate-2/6 block code over $\mathbb{H}(\mathbb{F}_3)$ [TS21, Example 1].

| H := QuaternionAlgebra over: $\mathbb{Z}/3$ invariants: #(-1 -1)

$\mathbb{H}(\mathbb{F}_3)$

| i := H i.
| M := H^6 span: {{1. 1. i. i. 1+i. 1+i}. {i. 1+i. 1+i. 1. 1. i}}.
| C := M asLinearCode

$[6, 2, 5]_{81}$

This code has 6561 codewords and minimum Hamming distance 5:

| C size

6561

| C minimumDistance

5

And we also compute its Hamming weight enumerator:

| C weightEnumerator

$$x^6 + 480xy^5 + 6080y^6$$

**Example 3.8.7.** (Number rings). The rings of integers of number fields are finitely generated $\mathbb{Z}$-modules, so we can construct them as finite algebras. For this we require an algorithm for computing an integral basis, currently implemented only for cyclotomic fields, quadratic fields and pure cubic fields.

We create the 5th cyclotomic field and its ring of integers:

| F := CyclotomicField new: 5

$$\mathbb{Q}(\zeta_5)$$

| R := F integers

$$\mathbb{Z}[\zeta_5]$$

The underlying module of $\mathbb{Z}[\zeta_5]$ is $\mathbb{Z}^4$. The integral basis is formed by powers of $\zeta_5$, so elements of $\mathbb{Z}[\zeta_5]$ are internally represented as $\mathbb{Z}$-linear combinations of powers of $\zeta_5$:

| R basis

$$(1, \zeta_5, \zeta_5^2, \zeta_5^3)$$

Now say we want to compute units of $R$ at random. (This is far from efficient, but it serves to illustrate some of the functionality currently available in the system.) We look for a polynomial in $F$ whose roots, if any, are units in $R$. We want the roots to be algebraic integers, so we make it monic with rational integer coefficients. And, because an algebraic integer is a unit if and only if its norm is a unit, we need the independent coefficient to be a unit of $\mathbb{Z}$.

```
P := ℤ polynomials.
x := P x.
[f := P atRandomMaxDegree: 3.
 f := x^5 + (f · x) + ℤ units atRandom.
 (roots := f rootsIn: F) isEmpty] whileTrue
```

We found the following units:

| roots

$$\{-\zeta_5^3 - \zeta_5^2 - 1, \zeta_5^3 + \zeta_5^2\}$$

We take any of the roots and coerce it to the ring of integers:

| u := R ! roots atRandom

$$-\zeta_5^3 - \zeta_5^2 - 1$$

**Example 3.8.8.** (Zero-dimensional affine algebras). We construct number fields, function fields and Galois fields as quotients of polynomial rings. However, since they are finite-dimensional vector spaces, we can construct them as finite algebras too.

| A := GaussianField new

$$\mathbb{Q}(i)$$

Although $A$ prints as $\mathbb{Q}(i)$, it is actually the affine algebra $\mathbb{Q}[x]/\langle x^2 + 1 \rangle$.

| F := A asFiniteAlgebra

$$\mathbb{Q}^2$$

This prints as $\mathbb{Q}^2$ but it's not really a vector space, it is a finite algebra with underlying vector space $\mathbb{Q}^2$, equipped with maps that allow coercion of elements from and to the affine algebra $A$. While elements of $A$ are residues of polynomials modulo $x^2 + 1$, elements of $F$ are represented internally as vectors of the underlying vector space $\mathbb{Q}^2$.

In general, from any zero-dimensional (that is, of Krull dimension 0) affine algebra over a field we can construct an isomorphic finite algebra.

## 4  Implementation Details and Performance Considerations

Tuples, matrices and univariate polynomials, often over small prime fields $\mathbb{F}_p$, lie at the bottom of most computations. It is desirable to optimize basic operations with these objects because they lead to performance improvements on the whole system.

Tuples, matrices and univariate polynomials are internally represented as arrays of coefficients, and delegate basic operations to their arrays of coefficients.[14] We have a single implementation of tuples, matrices and univariate polynomials independent of the coefficient ring, and different implementations of arrays for different types of coefficient rings. This design allows optimizations without impacting or limiting the mathematical expressiveness of the system.

We implement specialized arrays over $\mathbb{Z}/m\mathbb{Z}$ for small $m$ that store elements in a memory-efficient way. For $m = 2$, we store each element as a single bit in an array of 32-bit words. In this way, a polynomial of degree $n$ in $\mathbb{F}_2[x]$ or a tuple in $\mathbb{F}_2^n$ use only about $n$ bits of memory, and a dense $n \times m$ matrix over $\mathbb{F}_2$ uses about $nm$ bits of memory (plus a small additive constant of a few bytes). For $m \leq 256$ we store elements as bytes, and for $m \leq 2^{32}$ we store elements as 32-bit words.

Arrays over Galois rings $\mathrm{GR}(p^k, r)$ are internally represented as arrays over the base local ring $\mathbb{Z}/p^k\mathbb{Z}$, storing each element as $r$ consecutive elements of the base ring. In this way we take advantage of optimizations for arrays over the base ring $\mathbb{Z}/p^k\mathbb{Z}$ when $p^k$ is small. In particular, arrays over Galois fields $\mathbb{F}_{p^r} = \mathrm{GR}(p, r)$ are stored as arrays over the base prime field $\mathbb{F}_p = \mathbb{Z}/p\mathbb{Z}$. For example, a polynomial of degree $n$ in $\mathbb{F}_{2^r}[x]$ is stored in about $nr$ bits of memory.

This approach not only saves memory and enables the possibility of high-performance primitives in C, but also limits the number of instances of Smalltalk objects created during computations, and prevents stressing the garbage collector. Otherwise, for example a $1000 \times 1000$ matrix would have references to 1M instances (one for each coefficient), and performing simple arithmetic operations with big matrices would result in the instantiation of lots of small objects that would have to be garbage collected soon afterwards. It would be like if every pixel in the screen was a Smalltalk object, instead of using one Bitmap for the whole screen.

## 5  Future Work

During the first stage of development we focused on building general and mathematically rigorous foundations. In the following stage we plan to focus on improving performance and extending functionality.

In order to improve performance we plan to implement more efficient algorithms for some fundamental tasks such as computing determinants and the Hermite normal form over $\mathbb{Z}$, and implement high-performance primitives in C for basic operations with tuples, matrices and univariate polynomials over $\mathbb{Z}/m\mathbb{Z}$ with small $m$.

We also plan to extend the functionality of number and function fields and their orders, affine and finite algebras, schemes, permutation groups and linear groups. And we'd like to extend the computation of Gröbner bases to some noncommutative algebras, starting with algebras of solvable type [GP07, § 1.9] that include Weyl algebras, exterior algebras, and instances of quantum groups.

---

[14]More precisely, arithmetic operations are delegated to the arrays of coefficients, but higher level operations like the syzygies and lifting problems on matrices and polynomial factorization are delegated to the coefficient rings.

Furthermore, we plan to continue developing the machinery of homological algebra in terms of constructive category theory [Pos19].

## 6 Acknowledgements

We want to thank Leandro Caniglia for his continuous encouragement and advice, and for the suggestion to use standard LaTeX commands to input special Unicode characters; and Juan Vuletich, not only for making Cuis Smalltalk, but specially for his heroic work on TrueType fonts and Unicode support in Cuis that enabled the use mathematical notation in Smalltalk code.

We also want to express our gratitude to the FAST 2022 Workshop organizers for the opportunity to participate, and the anonymous reviewers for their corrections and very helpful suggestions.

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
