# OpenReview forum: "Arrows: a Computer Algebra System in Smalltalk"
_FAST.org.ar/2022/Workshop — FAST Smalltalk 2022 BestPaper_

### Official Review · Reviewer_9Ftw · 2022-10-30
**Review on ``Arrows: a Computer Algebra System in Smalltalk''**

**Rating:** 8
**Confidence:** 3

**Review:**

The work done by the author is admirable.
While similar packages (e.g. Singular, PARI, GAP, Magma) tend to have tens, sometimes hundreds of developers, having all this work done by only one developer is amazing.
I strongly recommend to accept the paper. My rating below is 8, but it could be higher (I have not enough knowledge to clearly distinguish between 50, 85 or 95 percentiles).

I further recommend the author to make the package publicly available.
Gaining users (and developers) is the only way for a project like this one to succeed and go beyond the early first steps.
In doing so, a comparison with other CAS could be included. Mainly, a list of the areas that the author identified other CAS did not enter yet.

### Changes proposed

The following is a list of minor remarks and changes that the author could address before publication.

1. Pg. 2, footnote 4: move it one sentence forward. That is, right after
   "Domains are objects of a category; that is to say, they are the domains and codomains of morphisms."
   The reason is that at the point it is inserted, the word "object" was not mentioned yet.

1. Pg. 4, section 3. In the requirements about `lift` and `colift` there could be an explanation of what
   should happen when `a` is a zero divisor and there is more than one solution to `ax = b` (or `xa = b`).
   This is even more important in pg. 5, in the definition of Computable Rings.
   Should `syzygies` and `lift` messages produce specific solutions? If so, which ones?

1. Pg. 5, the ring $M_n(R)$ is involutive for $R$ involutive (commutativity is not needed).

1. Pg. 6, footnote 12: when $R$ is not commutative, the notation $Hom(R^n, R^m)$ should specify which
   are the scalars expected to commute with maps. If all $m\times n$ matrices are considered as
   morphisms, then the scalars are the center of $R$. In that case, $Hom(R^n, R^m) = Hom_{Z(R)}(R^n, R^m)$
   is both a left and a right $R$ module, and then a right and a left $R^{op}$ module.

1. Pg. 7, Example 3.4.2. The meaning of the message `span` is not clear (to this referee). Neither the answer in

   ```
   H := (SymmetricGroup new: 6) span: #((6 1 2 3 4 5))
   ⟨(4 3 2 1 6 5)⟩
   ```


1. Pg. 10, Section 3.6, in the phrase
   "If $\bar A$ has n rows, the solution $\bar X$ we’re looking for is the matrix whose coefficients are the projections to R/I of the coefficients of the first n columns of X"
   the author might mean the coefficients of the first n **rows** of X.

1. Pg. 16, in "The following is an schematic sketch", change "an" to "a".

---

### Official Review · Reviewer_2XFc · 2022-10-30
**Arrows: a Computer Algebra System in Smalltalk**

**Rating:** 9
**Confidence:** 5

**Review:**

The computer algebra system (CAS) "Arrows" implements the category of finitely presented modules over a computable ring. This category is proven to be constructively Abelian and as such has many crucial applications. Computable rings are rings over which one can solve one-sided linear equations. The CAS "Arrows" supports several computable rings ( e.g., fields, Euclidean rings, polynomial rings) by implementing the respective algorithms for solving one-sided linear systems (by using the Gauss normal-form, Hermite normal-form, and Buchberger's Gröbner-basis algorithm, respectively).

The paper is well-written and the CAS seems to be well-designed.

Even if most (if not all) of the functionality is covered by other computer algebra systems, "Arrows" is still a very valuable contribution.

The future development of "Arrows" could profit from recent advances in the computer algebra community, where category theory is playing a more prominent role in the design of the algorithms. Here are two references:

https://arxiv.org/abs/1712.03492 (a vast generalization of finitely presented modules)
https://arxiv.org/abs/1908.04132 (homological algebra using involved category theory)

I strongly recommend the paper for publication.

Typos: Weil algebra -> Weyl algebra

---

### Official Review · Reviewer_funa · 2022-10-31
**Exciting system**

**Rating:** 10
**Confidence:** 3

**Review:**

Before moving to the actual review, I would like to say that this paper is the most exciting paper I have reviewed in the last year (around 20 articles).

## Paper Summary

This paper describes Arrows: a Computer Algebra System in Smalltalk. As it is said in the first sentence, "the goal of this project is to build an extensible environment where one can construct mathematical objects and operate with them, their morphisms, and their elements."

My understanding is that the system fulfilled its objectives wonderfully.
There are two main sections in this paper:

* Section 2 describes the core language of this paper: Domains, i.e., objects of a category, and morphisms. In this section, some special morphisms are introduced: for example, the message $A \to B$ representing the universal morphism from $A$ to $B$, or how to recover the (co)projections from (co)products.
* Section 3 describes the implementation and characteristics of some algebraic structures. To mention a few examples, there are rings (involutive, euclidean, computable, products and quotients), modules, localizations of rings, and sub-categories of R-algebras.
All around the place, the paper presents compelling examples, like Example 3.4.3, which describes how to create pullbacks and pushouts from module homomorphisms; Example 3.6.2, showing how to get Linear codes and its extension in Example 3.8.6, where the linear codes are computed over noncommutative rings.

The paper then finishes with a brief description of the implementation details focusing on the numerical methods.

## Strengths

- One feature that attracts me the most about Smalltalk is how objects can be livelily manipulated. This paper describes a system that excels at taking that viewpoint and applying it to the field of computer algebra. The result is a system that not only allows us to construct and build mathematical objects but also helps us learn and increase our understanding. I learned a lot from reading this paper.
- The paper presents many examples of what can be achieved using Arrows.
- Since the system was neatly designed, very powerful concepts can be accessed concisely. Consequently, some of the system's actual implementation and protocol are shown in the paper, interleaved with text. And the quality of that result was improved due to the system's use of Unicode. (On the flip side, it is not clear to me that readers unfamiliar with Smalltalk will understand that they are reading executable code and not just mathematical notation.)

## Weaknesses

- One common issue across many papers is that they need to identify their contributions (and what was previously done). This paper was not immune to that, but in an uncommon way: sometimes I had the impression that there are actual contributions to the state of the art, but the paper does not make that claim--may it be due to excessive humbleness? One example is the paragraph describing a computation similar to the Zassenhaus algorithm [before the end of page 8]. Is this expressing a novel generalization of said algorithm? A similar question appears regarding the computability of Euclidean rings as an extension of the computability of Euclidean domains.
- I looked for the project online, but I couldn't find it. Is it going to be available in an open-source Smalltalk?

## Further comments:

- I enjoy the coercive binary operator #! because a) it is very clear, and b) other Smalltalkers outside this project can benefit from using it.
- How does the system deal with infinite sequences/collections?